# Microbial adaptation to spaceflight is correlated with bacteriophage-encoded functions

Iris Irby [1,2] & Jared T. Broddrick [1] ✉

Evidence from the International Space Station suggests microbial populations are rapidly adapting to the spacecraft environment; however, the mechanism of this adaptation is not understood. Bacteriophages are prolific mediators of bacterial adaptation on Earth. Here we survey 245 genomes sequenced from bacterial strains isolated on the International Space Station for dormant (lysogenic) bacteriophages. Our analysis indicates phage-associated genes are significantly different between spaceflight strains and their terrestrial counterparts. In addition, we identify 283 complete prophages, those that could initiate bacterial lysis and infect additional hosts, of which 21% are novel. These prophage regions encode functions that correlate with increased persistence in extreme environments, such as spaceflight, to include antimicrobial resistance and virulence, DNA damage repair, and dormancy. Our results correlate microbial adaptation in spaceflight to bacteriophage-encoded functions that may impact human health in spaceflight.

Human space exploration is at a turning point. The push for long-duration, crewed missions to the moon and beyond, coupled with the emergence of commercial space tourism is redefining the risks associated with space travel. Adverse health effects due to host-microorganism interactions is acknowledged by the NASA Human Research Program, and currently this risk is accepted for short duration spaceflight. However, as flight durations increase and the population that experiences spaceflight expands beyond professional astronauts, it is of increasing importance to understand the interplay between microbial adaptation to the spaceflight environment (e.g., microgravity, ionizing radiation, altered host-microbe interactions, etc.) and human health.

Microorganisms on the International Space Station (ISS) are undergoing adaptation and host-mediated propagation throughout the environment[1–3]. Studies to date suggest microbes on the ISS, predominantly derived from the crew microbiome, are enriched in functions related to metal ion tolerance, dormancy, and antibiotic resistance[3,4]. A primary mechanism by which bacteria adapt to novel environments and stress is through genome plasticity, the ability to uptake exogenous genetic material[5]. The presence of temperate bacteriophages, or prophages, in bacterial genomes is one of the main factors that enable genome plasticity[6]. The entrance of a phage into a bacterial genome is performed through the process of lysogeny, forming a "dormant" prophage containing the entirety of the bacteriophage DNA that is not actively bactericidal, but can undergo replication into a mature virus. These prophages can remain dormant for extended periods of time, or switch into their active (lytic) state, typically after intense DNA damage. Lytic phages can become bactericidal and infect other bacterial cells, taking either their lytic or lysogenic state in their new host[7].

Prophages dramatically alter their host's physiology. The presence of prophages can disrupt and silence bacterial genes, and they can cause chromosomal rearrangements, leading to changes in gene expression in their bacterial host. Prophages can confer host immunity to other bacteriophages, preventing lytic attack[8]. During the transition to the lytic state, prophages can copy adjacent bacterial genes, integrating them into the viral genome. This genetic material is transferred to new bacterial hosts[9]. Examples of phage transduced functions include virulence and antimicrobial resistance genes, toxins, effector proteins, and regulatory proteins[8].

[1]Space Biosciences Research Branch, NASA Ames Research Center, Moffett Field, CA, USA. [2]School of Biological Sciences, Georgia Institute of Technology, Atlanta, GA, USA. ✉e-mail: jared.t.broddrick@nasa.gov

Prophages also affect microbial populations at the community level. The introduction of bacteriophages into soil samples significantly changed the microbial composition and diversity of the soil microbial community[10]. In aquatic environments, bacteriophage lyse 20–40% of bacterial biomass per day[11]. In mouse model studies, both temperate and lytic phages influence the host gut microbiome, with lytic phages leading to a more beneficial community, and temperate phages leading to dysbiosis[12].

The role of bacteriophage in the adaptation of bacteria to the spaceflight environment is unknown. Previous studies have highlighted the importance of microbial communities in spaceflight, with changes noted in the composition of both the gut and skin microbiomes of astronauts[13]. While other studies have noted the presence of prophages in a single bacteria species isolated from the ISS[14], in this work we show a comprehensive analysis of the prevalence and characterization of prophage that inhabit the spacecraft-built environment.

## Results

### Genome summary and sampling cadence and locations

The International Space Station (ISS) is a unique built environment consisting of 388 cubic meters of habitable space. Crew size has ranged from 3 to 10 over the 22 years of continuous habitation. Surfaces within the occupied modules have been continuously sampled and analyzed for microbial populations to understand the microbiome of the built spacecraft environment. We set out to analyze the genomes that resulted from these experiments to assess the role of bacteriophage in the spacecraft-built environment.

We identified 10 bacterial strains of interest (see Methods): *Acinetobacter pittii*, *Bacillus amyloliquifaecins*, *Bacillus thuringiensis*, *Enterobacter bugandensis*, *Klebsiella quasipneumoniae*, *Micrococcus luteus*, *Paenibacillus polymyxa*, *Pseudomonas fulva*, *Staphylococcus epidermidis*, and *Staphylococcus saprophyticus*. These 10 species represent isolates from sampling campaigns spanning from 2006 to 2018 from eight locations within the ISS (Fig. 1) and a Crew Resupply Vehicle while it was docked with the ISS. Overall, 245 spaceflight genomes and metagenome-assembled genomes[15] were included in our analysis. A full table with links to genomes and data sources can be found in Supplementary data 1.

### Prophage region identification and comparison

We extracted prophage regions from flight genomes for each species using PHASTER[16,17] and summed the prophage regions identified (categorized by PHASTER as complete, questionable, or incomplete), as these region numbers are proportional to the number of prophages that, at some point, lysogenized and entered the bacterial genome. For comparison, we obtained all terrestrial, complete genomes for these species from NCBI[18] and determined the number of prophage regions for these genomes (NCBI accessions can be found in Supplementary data 1). Average number of prophage regions in the flight genomes ranged from 0.75 per genome for *M. luteus* to 7 for *P. fulva* (Fig. 2). For *M. luteus*, 3 of the 8 flight isolates had two possible incomplete prophage regions per genome, but the PHASTER results suggested they were low confidence annotations and prophage analysis by PhiSpy[19], which employs more stringent criteria, did not identify any prophage regions. Thus, *M. luteus* was removed from further analysis. Of the nine remaining species, all had multiple incomplete prophage regions and at least one complete phage, which, in theory, could convert to a lytic phenotype. In addition, five species presented with poly-lysogeny defined as containing multiple complete prophages. There were no universal trends in the number of prophage regions in the flight genomes versus the terrestrial isolates (Fig. 2), with four species having significantly more prophage in flight genomes, four with more in terrestrial, and one that was insignificant (Fig. 2, Mann–Whitney, two-sided, $p < 0.05$).

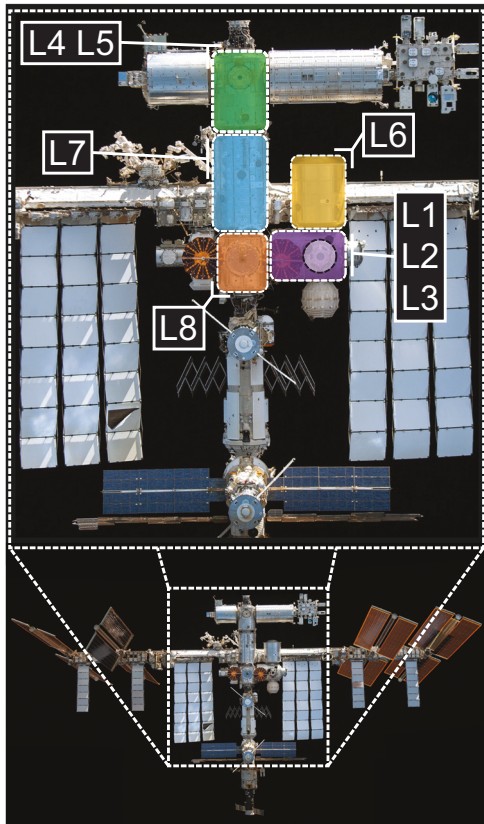

**Fig. 1 | Sampling locations on the International Space Station.** Purple shaded area: Node 3; sampling locations: Cupola (L1), Waste and Hygiene Compartment (L2), Advanced Resistive Exercise Device (ARED) (L3), Green shaded region: Node 1; sampling locations: dining table (L4), overhead 4 (L5), Yellow shaded region: Permanent Multipurpose Module; sampling locations: (L6), Blue shaded region: US Lab sampling locations: Lab Overhead 3 (L7), Orange shaded region: Node 2; sampling locations: crew quarters (L8). Images courtesy of NASA.

Due to the consistency in flight isolate prophage numbers for most of the species investigated (Fig. 2), coupled with the sampling cadence on the ISS (Supplementary data 1), we explored the genomic diversity between the flight isolates and the terrestrial strains. We used a pangenome approach to compare the similarity within the flight strains for each species, compared to the pangenome for all strains of the same species (all genomes in NCBI with an Assembly level at or above scaffold). We defined the core pangenome as genes present in at least 90% of the genomes, with the remaining genes defined as the accessory pangenome. We looked at two metrics for our analysis. First, we looked at the ratio in the size of the flight-strain pangenome versus the all-strain pangenome, working from the assumption that the more similar the flight strains, the larger the flight-strain core genome. Second, we took the fraction of the flight-strain core pangenome versus the total flight-strain pangenome (core plus accessory), as the smaller the accessory genome the less genetic diversity between the ISS strains. For high-confidence prophage regions (prophage calls consistent between PHASTER and PhiSpy algorithms), we removed the prophage regions from the genomes (replaced with N) to capture phage-independent genome variability.

Flight strains for four of nine species (*B. amyloliquifaeciens*, *K. quasipneumoniae*, *S. epidermidis*, and *S. saprophyticus*) had a core pangenome fraction of 0.90 or less (Supplementary data 1). The accessory pangenome of the remaining five species was 3% or less of the total pangenome, suggesting a high level of genetic similarity between the flight strains for these species. The ratio of the flight strains' core pangenome versus the total species core pangenome was

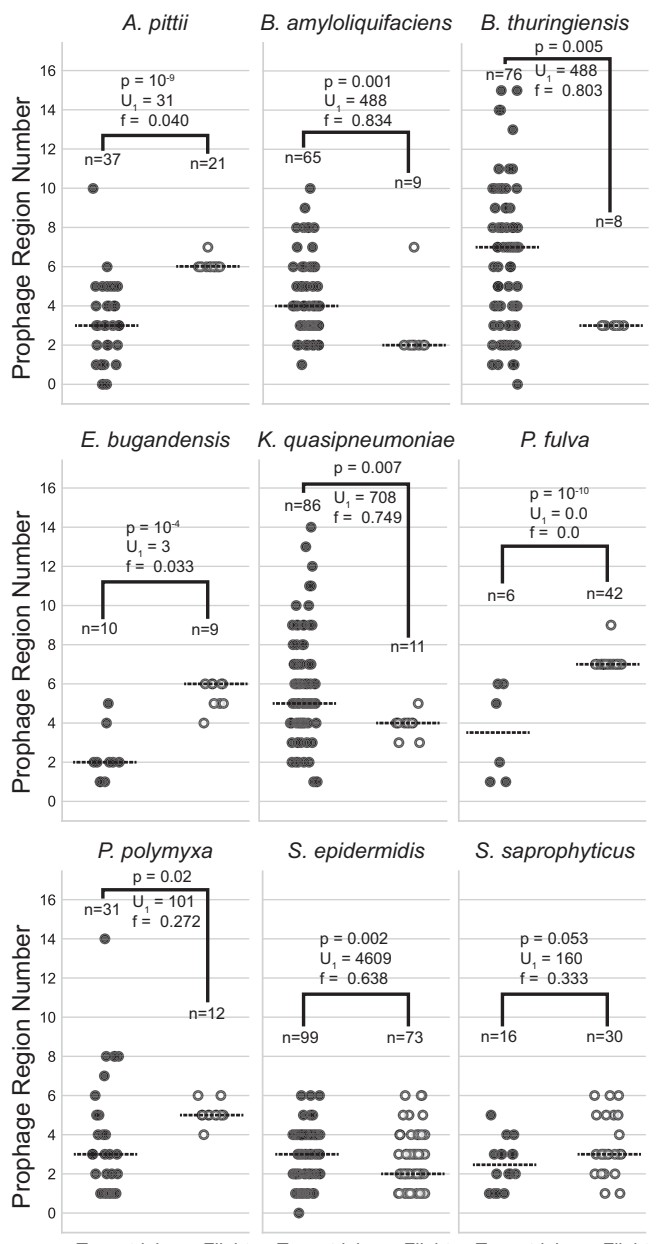

**Fig. 2 | ISS (flight) versus terrestrial prophage counts.** The median is indicated by a dashed horizontal line for each condition (flight or terrestrial) for each species. The number of genomes in the flight and terrestrial categories are indicated adjacent to each group of data. Two-tailed Mann–Whitney $p$ value, for $H_0$ $n_{flight} = n_{terrestrial}$, and the U-statistic and Common Language Effect Size (f) for the terrestrial condition are shown. Source data are provided as a Source Data file.

larger for all species, as expected for strains continuously sampled from the same environment. However, the magnitude of the ratio mirrored the change in the core pangenome fraction with the *K. quasipneumoniae*, *S. epidermidis*, and *S. saprophyticus* flight strains having similarly sized core pangenome as the all-strain pangenome (1.06, 1.01, and 1.03, respectively). *S. epidermidis* had the most longitudinal sampling cadence with 30 separate samples taken over 12 years (Supplementary data 1), which correlated with the highest level of genetic diversity between flight genomes (flight versus all core ratio of 1.01 and a flight ISS core fraction of 0.54). However, there were no clear trends in the pangenome comparison versus sampling cadence for the remaining eight species. For example, the 21 flight strains for *A. pittii* were sampled across three campaigns, two of which were 3 years apart.

Still, these genomes showed the highest pangenome core ratio (1.21) and a flight core versus total pangenome fraction of 0.97, with only 89 genes in the accessory pangenome. *S. saprophyticus'* 29 flight genomes came from a similar sampling cadence while having the second most genetic diversity among flight strains. The full genome diversity comparison can be found in Supplementary data 1.

We then applied the pangenome approach to determine if prophages contribute to variation between flight and terrestrial strains of the same species. Again, using all genomes in NCBI with an Assembly level at or above scaffold, but not removing the high confidence prophage regions, we identified Clusters of Orthologous Groups (COG) functions that were significantly different ($q$ value < 0.05) between the flight and terrestrial genomes. For both flight and terrestrial genomes, there were significant differences in gene clusters associated with the phage-associated COG Mobilome (Fig. 3). Terrestrial genomes were enriched for transposases across multiple species, including *A. pittii*, *B. thuringiensis*, *K. quasipneumoniae*, and *S. epidermidis*, compared to spaceflight genomes. Spaceflight genomes were enriched for Mu-like phage structural proteins in *A. pittii*, *E. bugandensis*, and *P. fulva* and enriched for general phage structural proteins in *A. pittii*, *E. bugandensis*, *K. quasipneumoniae*, *P. fulva*, and *S. epidermidis*. Only *S. saprophyticus* contained prophages that were not significantly different between flight and terrestrial genomes. In addition, *A. pittii*, *B. amyloliquifaecins*, *B. thuringiensis*, *E. bugandensis*, and *P. fulva* indicated unique phage-associated gene clusters in both flight and terrestrial genomes suggesting a possible divergence in prophage regions resulting from spaceflight.

**Complete prophage classification**
To investigate the novelty of the flight prophages, we analyzed the complete prophage regions in comparison to known phage. In total, 283 complete prophage regions were identified in the 9 bacterial species' flight genomes and all 283 were consistent between PHASTER and PhiSpy, suggesting high confidence prophage annotations. We clustered these complete prophages into 71 bins of >90% similarity, and determined the hypothetical phage family of each cluster by NCBI nucleotide BLAST search [Megablast[20]] of viruses in the RefSeq Genome Database[21] and the IMG Viral (VR) Database[22]. The IMG Viral database was the more comprehensive of the two with an average query identity score (nucleotide identity multiplied by alignment percentage) of 55%, compared to 28% for the RefSeq viral genomes. Of the 71 clusters, 11 were unclassified (15%) based on having a total query identity of less than 40% with any phage in the RefSeq or IMG VR Databases. Of the remaining 60 clusters, all complete prophages were found to be tailed phages of class Caudoviricetes (Supplementary data 1). Of these 71 intact prophage clusters, 21 could be classified as novel species and 26 as novel genera based on current suggested cutoffs of 95% query identity for species and 70% for genera[23] (Fig. 4a).

Next, we determined the homology of the complete prophage regions to bacterial genomes in the RefSeq database. As lysogenic prophage regions are incorporated into the bacterial genome, homologous regions may be present in terrestrial genomes in the bacterial database. The average query identity increased from 55% in the viral database to 80% with the bacterial database. Using the same cutoffs as above, 21 of the 71 clusters would be considered novel at the 95% query coverage cutoff and 16 would be considered novel at the 70% cutoff (Fig. 4b). Of the nine species, only *B. thuringiensis* lacked a novel complete prophage at the 70% cutoff (hereafter: non-redundant prophage region), suggesting unique prophage-associated genomic content was broadly observed in the flight genomes (Table 1).

Finally, we attempted to elicit a lytic phenotype from the flight strains. Seven isolates from four different BSL1 species [*B. amyloliquifaeciens* (IIF7SC-B1 and IIF7SW-P4), *P. fulva* (F8_7S_9B), *P. polymyxa* (IIF5SW-B4 and IIF8SW-P3), *and S. saprophyticus* (F6_7S_P5 and F5_7S_P13)] were provided by the NASA Jet Propulsion Laboratory. As

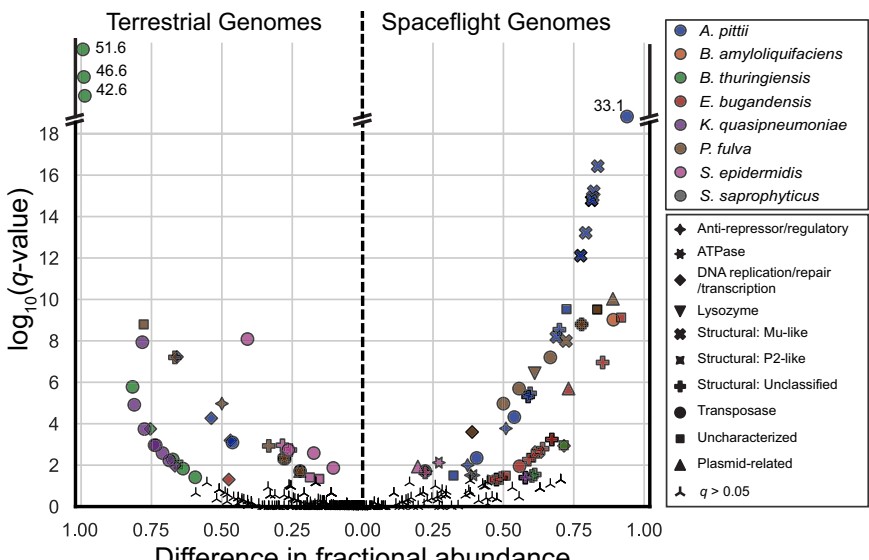

**Fig. 3 | Mobilome-associated Clusters of Orthologous Groups (COG) functions significantly different in either flight or terrestrial genomes [*q* value < 0.05, two-tailed Rao score test *p* value corrected for multiple testing (see Methods)].** General categories of phage genes are given different shapes, while the different species are show in different colors. $\log_{10}$(q-values) > 18 have their values shown adjacent to their respective data point. Source data are provided as a Source Data file.

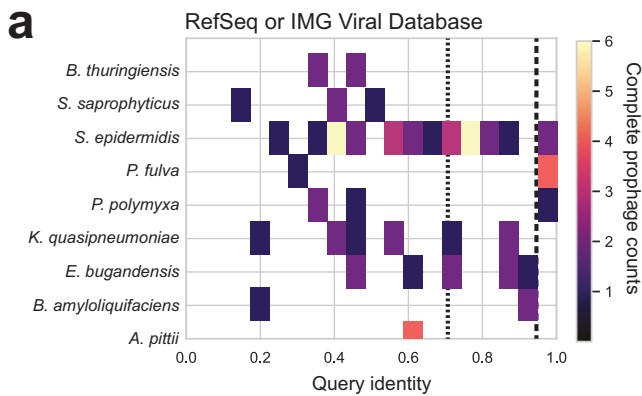

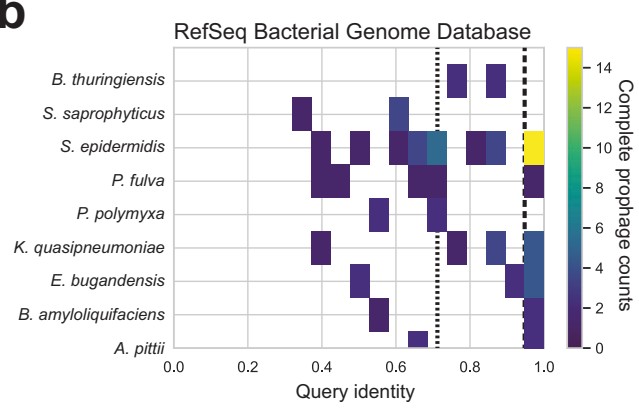

**Fig. 4 | Homology of complete prophage regions versus viral and genome databases. a** Query identity (nucleotide query coverage fraction × by identity fraction) for complete prophage regions in each species queried against the RefSeq and IMG viral databases. The higher value between the two databases was plotted. **b** Query identity (nucleotide query coverage fraction × by identity fraction) for complete prophage regions in each species queried against the RefSeq bacteria database. Vertical dashed lines indicate the 0.95 and 0.70 query identity cutoffs. Source data are provided as a Source Data file.

the phage induction mechanisms have not yet been explored in these species, we subjected them to two common DNA damaging stimuli, mitomycin C and UV exposure. For all strains, log-phase cultures treated with mitomycin C (500, 100, 20, 4, and 0.8 ng mL⁻¹) showed a dose-response decrease in culture viability, as measured by OD600, ~8 h post-mitomycin C addition (Supplementary Fig. 1). In contrast, cell viability was more variable following UV-C radiation (254 nm; 5, 25, 50, 100, 200 J m⁻²). Both *S. saprophyticus* species showed dose-dependent effect on the growth curve (b–c). This was similar to both *B. amyloli-quifaeciens* isolates for the initial 12 h of growth, after which the growth curves were indistinguishable between treatments, which coincided with the formation of a biofilm (Supplementary Fig. 2d, e). The data for *P. polymyxa* was inconclusive as this species' biofilm growth pheno-type made it difficult to standardize culture starting densities and resulted in non-uniform light attenuation during OD600 determina-tion (Supplementary Fig. 2f, g). *P. fulva* growth curves showed very little change in response to UV expose, which may suggest enhanced UV resistance in this ISS isolate (Supplementary Fig. 2a).

For all species with more than one isolate, the complete pro-phages were identical in both strains, preventing us from testing for bacteriophage infection via the soft-agar overlay method. Instead, we attempted to purify phage particles post-mitomycin C addition via filtration (0.22 μm) and precipitation (8% PEG8000, 1 M NaCl). While we were unable to recover phage particles, our attempt was cursory, and a more comprehensive exploration of phage induction in these isolates is needed.

**Long-read assemblies provide insight into mobile genetic elements**

We explored the contribution of genome assembly level on the prophage predictions by both the PHASTER and PhiSpy algorithms. From the seven isolates we sequenced two strains, *P. fulva* F8_7S_9B and *S. saprophyticus* F5_7S_P13, via Nanopore MinION. *P. fulva* was chosen as all the spaceflight strains are poly-lysogenic with a non-redundant prophage region in each strain (Cluster 52, Table 1). *S. saprophyticus* F5_7S_P13 is a member of Cluster 41 containing a non-redundant prophage region (Table 1).

Assemblies for both species resulted in a single contig for the bacterial chromosome. *S. saprophyticus* F5_7S_P13 contained three

**Table 1 | Non-redundant prophage regions in spaceflight isolates**

| Species | Cluster | Representative Phage | Genomes in Cluster | Size (kb) | Family | Query Identity[a] | | |
|---|---|---|---|---|---|---|---|---|
| | | | | | | NCBI Virus | IMG Virus | NCBI Bacteria |
| *A. pittii* | 2 | IIF1SW-P2 sequence:120779-155538 | 13 | 34.8 | Caudoviricetes | 0% | 65% | 59% |
| | 3 | IIF1SW-P4 sequence:16202-50961 | 8 | 34.8 | Caudoviricetes | 0% | 65% | 59% |
| *B. amyloliquefaciens* | 6 | IIF7SW-B4 sequence:76-30790 | 2 | 30.7 | Caudoviricetes | 0% | 55% | 21% |
| *E. bugandensis* | 12 | IFACSW-P1 sequence:830462-877738 | 7 | 47.3 | Caudoviricetes | 11% | 52% | 47% |
| | 17 | IF2SW-F2 sequence:448858-496167 | 2 | 47.3 | Caudoviricetes | 11% | 51% | 47% |
| *K. quasipneumoniae* | 24 | IIIF3SW-P1 sequence:1181253-1232366 | 2 | 51.1 | Unclassified | 4% | 37% | 20% |
| *P. fulva* | 52 | F8_8S_9P sequence:173154-179665 | 1 | 6.5 | Caudoviricetes | 0% | 56% | 33% |
| *P. polymyxa* | 26 | IIF5SW-B4 sequence:979867-1033318 | 5 | 53.5 | Caudoviricetes | 0% | 56% | 33% |
| | 28 | IIF2*SW-P4 sequence:600724-654175 | 2 | 53.5 | Caudoviricetes | 0% | 66% | 29% |
| *S. epidermidis* | 34 | IF4SW-P4 sequence:652269-693911 | 2 | 41.6 | Caudoviricetes | 20% | 58% | 55% |
| | 38 | s34 sequence:116777-162391 | 2 | 45.6 | Caudoviricetes | 16% | 68% | 41% |
| | 62 | s51 sequence:616-49418 | 1 | 48.8 | Caudoviricetes | 11% | 65% | 54% |
| | 69 | s17 sequence:35-44746 | 1 | 44.7 | Unclassified | 29% | 40% | 39% |
| | 74 | F6_3S_P_7 sequence:72993-118974 | 1 | 46 | Caudoviricetes | 14% | 48% | 45% |
| *S. saprophyticus* | 41 | F6_7S_P_5 sequence:592884-635056 | 2 | 42.2 | Caudoviricetes | 43% | 63% | 25% |
| | 42 | F5_7S_P5C sequence:174-42331 | 5 | 42.2 | Caudoviricetes | 43% | 63% | 25% |
| | 70 | s43 sequence:757637-811609 | 1 | 54 | Unclassified | 5% | 32% | 12% |
| | 71 | F5_7S_P11B sequence:38923-82523 | 1 | 43.6 | Caudoviricetes | 50% | 61% | 24% |

[a]NCBI Megablast query coverage × alignment.

plasmids of 30, 33 and 36 kb in length. For this species, all five of the short-read assembly prophage regions mapped to the long-read assembly. However, one short-read incomplete prophage region was missing in the long-read assembly PHASTER analysis, despite the entire prophage region mapping to the long-read assembly (Supplementary data 1). This prophage region was present in the PhiSpy prophage calls for the long-read assembly. In addition, the long-read prophage analysis suggested the complete prophage from this species was truncated in the short-read assembly (42.1 kb versus 56.2 kb). Interestingly, the incomplete prophage #5 from the short-read prophage analysis (Accession: NZ_JAFDPF010000007.1, sequence: 10127 to 19977) was located on the 36 kb plasmid.

The prophage analysis for *P. fulva* F8_7S_9B suggested the short-read assembly led to fragmented prophage analysis. The long-read assembly identified 9 prophage regions, compared to 7 in the short-read assembly, with one short-read incomplete prophage region missing in the long-read assembly analysis (Supplementary data 1). Similar to *S. saprophyticus*, the entire prophage region mapped to the long-read assembly with no clear reason for its exclusion from the PHASTER prophage calls. This prophage region was not identified by PhiSpy in the long-read analysis. Long-read prophages #1, 5 and 6 were absent in the short-read analysis due to the prophage regions being split across separate contigs in the short-read assembly. Of the two complete prophages in *P. fulva* F8_7S_9B, a 62.9 kb long-read prophage region was split into two short-read prophage regions due to lying on two adjacent contigs. The other complete prophage region was annotated as a single 47.6 kb region in the long-read assembly but as two prophage regions in the short-read assembly (intact: 35.6 kb and incomplete: 12.1 kb region). The long-read assemblies can be found on NASA's Open Science Data Repository (OSD-582, DOI: 10.26030/v6s6-w170) and NCBI (see Data Availability Statement).

**Functional analysis of prophage region-encoded proteins**

We selected a representative strain from each species and curated the prophage-encoded bacterial gene annotations to assess the functional contribution of phages to their hosts. Of the functionally annotated content, 46% was uncharacterized or hypothetical proteins. Of the remaining content, 20% was annotated as related to phage processes

such as structural proteins (e.g., capsid, tail, etc.), lysogeny, and phage replication (e.g., integrases, terminases). The remaining content was annotated as microbial proteins, conferring a variety of capabilities to the host species (Fig. 5). The total number of prophage-encoded bacterial proteins that could be ascribed a function ranged from ~60 for both *Staphylococcus* species to a high of 102 for *E. bugandensis* (707 total for all species). The most prominent functional category contained proteins related to Signal Transduction and Transcriptional Regulation (104 proteins across all species), followed by DNA Repair and Replication (91 proteins), and Cell Wall Biogenesis and Degradation (65 proteins). To investigate the subset of prophage-encoded proteins that provide unique capabilities to their host, we extracted prophage-encoded annotations that lacked functional homology to host genome-encoded proteins (Fig. 5). We computed a conservation score for each category defined as the fraction of prophage-encoded proteins that were functionally unique. While only consisting of five proteins across three species, Secretion Systems had the highest conservation score of 1.0 (100% of the prophage-encoded proteins conferred a unique capability to their host). Translation, Ribosomal Structure and Biogenesis, as well as Lipid and Fatty Acid Metabolism also had high conservation scores (0.9 for both categories). The prominent prophage-encoded functional categories of Signal Transduction and Transcriptional Regulation, DNA Repair and Replication, and Cell Wall Biogenesis and Degradation had conservation scores of 0.6, 0.51, and 0.48, respectively.

Many of the functions conferred by prophage-encoded proteins are correlated to increased fitness in competitive environments. All nine species had putative virulence related genes encoded on prophage regions, with 45% of these conferring unique functions to their hosts. Based on the assessment of antibiotic resistance genes using ABRicate, *P. polymyxa* prophage contained an aminoglycoside resistance gene (*aadK*), and *S. epidermidis* prophage contained two confirmed antibiotic resistance genes, a macrolide resistance gene [*mph(C)*] and an efflux pump [*msr(A)*]. Approximately 33% of the putative AMR proteins were possible resistance pumps for macromolecules. *A. pittii*, *B. amyloliquifaeciens*, and *E. bugandensis* all contained prophage-encoded pyocin-related defense proteins. Both *Staphylococcus* species contained prophage-encoded pathogenicity

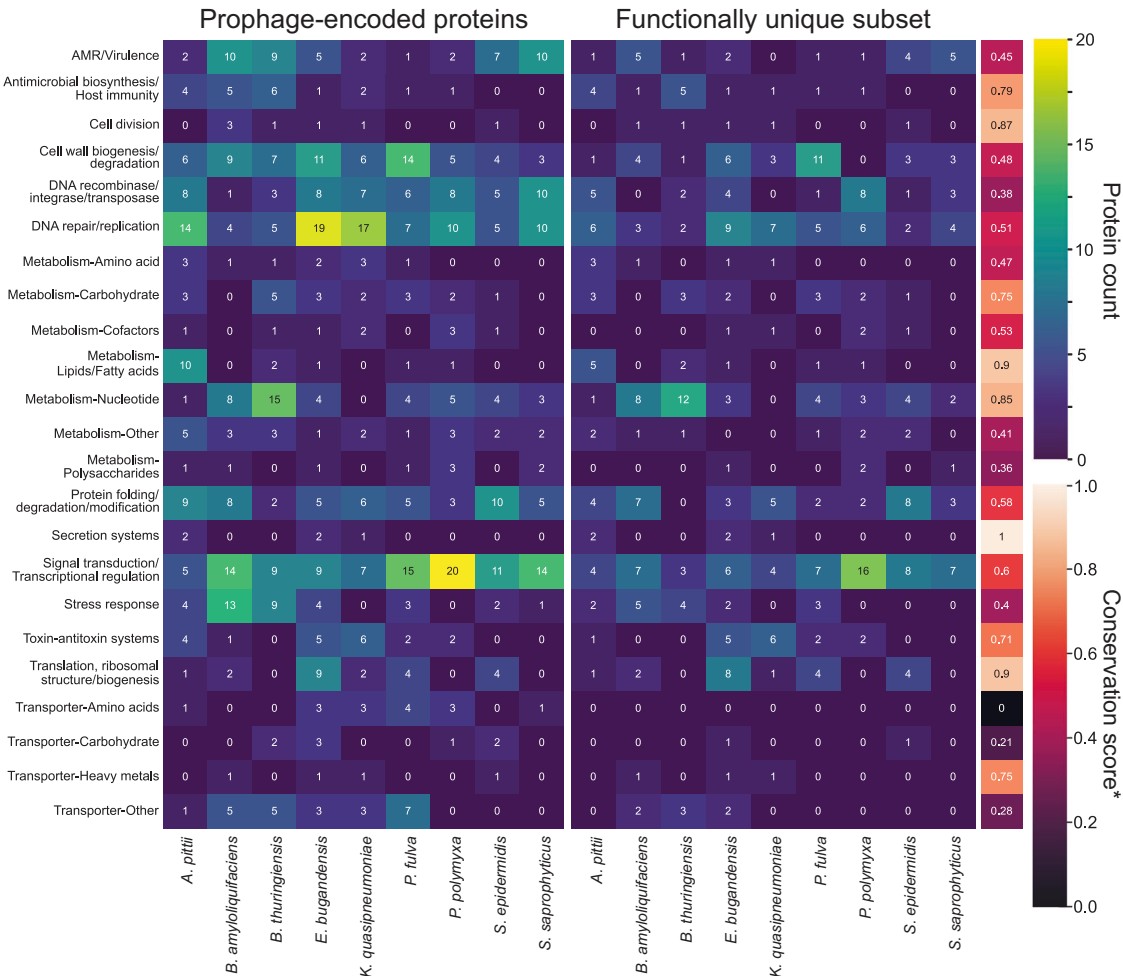

**Fig. 5 | Species-specific prophage-encoded protein functions.** Left panel: all prophage encoded proteins in each species. Right panel: the subset of prophage-encoded proteins that confer a unique function to their host. *Conservation score: mean of the functionally unique protein count divided by the total prophage protein count for each functional category. Phage-related proteins and uncharacterized/hypothetical proteins were excluded from the figure.

island genes and enzymes associated with putative virulence or antibiotic resistance functions (Supplementary data 1). *A. pittii*, *B. amyloliquifaeciens*, and *B. thuringiensis* contained prophage-encoded enzymes for deoxynucleotide sugar biosynthesis found in both antimicrobial biosynthetic pathways, such as streptomycin, as well as polyketide biosynthesis related to pathogenicity and antimicrobial activity[24]. These enzymes were unique to the prophage regions, with no homologs found in the non-prophage host genomic content.

Additional prophage-encoded gene products may confer fitness in the built spacecraft environment. Stress response proteins included those associated with sporulation and hibernation and metal resistance genes and transporters. Both categories have been shown to be enriched in spacecraft-associated microorganisms[3]. Four species had prophage-encoded LexA-related transcriptional regulators or repressors involved in the regulation of the SOS response in bacteria, which have recently been shown to be uniquely present in ISS strains of *A. pittii*[25]. This study also showed the presence of the RelBE toxin-antitoxin system in ISS *A. pittii* strains, which is also prophage-encoded.

## Discussion

Prophages are cited to be the largest factor associated with strain variation and diversity within terrestrial bacterial species[5,7], which holds in the spaceflight environment as phage-associated genes were significantly different in 8 of the 9 phage-containing species analyzed. Several spaceflight species contained genes related to the production

of Mu-like phage proteins compared to their terrestrial counterparts. Mu-like phages are unique in that they randomly integrate into the host genome, increasing mutation rates and host gene disruption[26]. Increased genetic diversity via Mu-like phage induced mutation may assist in microbial adaptation to the spaceflight environment and would explain the retention of these phage in flight strains. In comparison, terrestrial genomes contained transposases absent in the flight strains. Transposases are common in dense populations where there is rampant DNA sharing across a population of individuals[27]. The absence of these transposase regions may be related to the nutrient poor environment of the ISS. The presence of transposases in the terrestrial organisms may also be indicative of active mobile genetic elements, which we were unable to confirm for the spaceflight isolates.

Our analysis suggested a high number of novel prophages in the flight genomes. In total, we identified 283 complete prophages making up 71 unique prophage clusters, 18 of which were hypothesized as novel, and host-related genes making up over a third of the genes present in these prophage regions. While the discovery of novel prophages is not uncommon, with viral dark matter reported to be 70–95% of human associated viromes[28,29], a previous viral metagenomic study was able to classify the majority of the viruses in extreme environments[30].

Prophages are a primary mechanism of bacterial host gene transfer, to include antibiotic resistance and virulence genes[31]. We identified at least one putative antibiotic resistance or virulence host-related gene within the prophage regions of all species studied. In

comparison, <12% of highly antibiotic resistant ESKAPE pathogens carry a prophage-encoded antibiotic resistance gene[32]. In addition, lysogens carrying antibiotic resistance genes have been shown to provide a fitness benefit for their host[31,33]. There are several reports of spaceflight-induced increases in antibiotic tolerance or increased antibiotic resistance gene clusters[14,25,34–36]. With our identification of confirmed antibiotic genes within prophage regions of *P. polymyxa* and *S. epidermidis*, as well as the conservation of prophage regions within members of the same species, we describe one possible mechanism for this increased antibiotic tolerance.

We also identified categories of genes related to host immunity, stress response, and DNA repair. The presence of prophages within a bacterial host can provide immunity to attack from other bacteriophages or phage-like pyocins by inhibiting lytic phage replication, or repressing other phages[37,38]. We identified several host immunity genes in the flight prophages that could mediate phage and pyocin resistance. In addition, both *Bacillus* species contained prophage-encoded sporulation genes, related to persistence in harsh terrestrial environments[39,40] and spaceflight[3]. Prophage-encoded DNA repair genes were present across several species, notably those activated in response to radiation. UV radiation induced activation of lytic phage is well known; however, the ISS structure shields the interior from UV radiation. The presence of DNA-repair genes may play a role in the protection of the host genome from galactic cosmic rays or general DNA damage from desiccation in the built environment[41].

The association between genomic features enriched in ISS isolates and prophage regions suggest phage are important for adaptation to the spaceflight environment. However, these are correlations, and experimentation is required to directly connect prophage-mediated genetic plasticity with persistence in the built spacecraft environment. In addition, the opportunity for phage infection of microbial strains after they leave the crew microbiome is likely limited. Thus, an important line of inquiry is to assess the lytic-lysogenic traits of microorganisms while they are still host-associated.

## Methods

### Flight and terrestrial genomes
Candidate flight species to use in our analysis were determined from literature[42–44]. Isolates from the ISS were identified in the NCBI databases[18] by searching the BioSample database for "geo_loc_name=USA: International Space Station"[attr] AND "SPECIES NAME". These hits were mapped back to the NCBI Assembly database and the flight genomes were validated and downloaded. The terrestrial genomes were downloaded from the Assembly database with the following conditions: Latest RefSeq only, Exclude anomalous and Taxonomy Check Status OK. The Assembly Level differed based on analysis with Scaffold and above used for pangenome analysis and Complete genome was used for prophage identification. The Assembly Level for all ISS flight genomes were either Scaffold or Contig. Metagenomes were downloaded from the NASA Open Science Data Repository OSD-69 (DOI: 10.26030/cary-q426)[45]. A full list of genomes, reference links and sampling locations can be found in Supplementary data 1.

### Prophage identification and classification
Prophage regions for both ISS and terrestrial isolates were identified with PHASTER[17]. Results were separated into complete, questionable, and incomplete prophage regions and the prophage regions in fasta format were downloaded from the PHASTER web-based server. In addition, flight and terrestrial genomes, downloaded from NCBI using E-utilities (v 1.77) in GenBank format, were also searched for prophage regions using PhiSpy [v4.2.20,[19]] using default parameters. PhiSpy prophage regions consistent with PHASTER prophage regions were analyzed using ABRicate (v1.0.0, Seemann T, Abricate, Github https://github.com/tseemann/abricate) with all associated databases[46–53] to identify antibiotic resistance genes in prophage regions. Fasta files of

the bacterial genomes without prophages were exported and used for the flight isolate genome diversity analysis. Prophage regions from ISS strains were queried against the Virulence Factor Database (accessed April 2023)[53] to identify putative virulence factors using NCBI Blast[20]. Statistical differences in prophage number were determined using a two-tailed Mann–Whitney U rank test, correcting for ties, as implemented in SciPy [v1.7.3,[54]]. The effect size was computed as the common language effect size f value where $f = \frac{U}{n_1 n_2}$. The U-statistic and f value were reported for the terrestrial condition. Prophage regions identified as complete were clustered using MeShClust [v3.0,[55]] with a cut off of 90% nucleotide identity. The nucleotide sequence labeled as the center of each cluster was assessed for homology with known viral and bacterial sequences in the NCBI RefSeq genome database[21] using BLAST [Megablast[20], accessed November 2022] and high confidence genomes in the IMG/VR database [v4.1, accessed July 2023[22]]. Phage classification was determined based on the taxonomic similarity to the best hit in the viral database (query coverage percentage multiplied by percent identity > 40%).

### Pangenome analysis
Pangenome analysis of the flight and terrestrial genomes from NCBI and the NASA Open Science Data Repository[56] was completed using the pangenomics workflow[57–59] in anvi'o [v7.1[60]]. For *Bacillus thuringiensis, Klebsiella quasipneumoniae*, and *Staphylococcus epidermidis*, the terrestrial genomes were subsampled into sets of 300 genomes and the pangenome workflow run separately on each subset (2 sets for *B. thuringiensis* and *K. quasipneumoniae* each, and 3 sets for *S. epidermidis*). For the ISS flight genome diversity analysis, the non-prophage containing bacterial genomes output from PhiSpy was used for analysis. Outputs from the anvio-summarize function were parsed for the number of genomes in which each gene was present. The core pangenome was defined as genes present in at least 90% of the genomes (or for ISS species with less than 10 genomes, the cutoff was n-1 genomes, where n is the number of available genomes). The accessory pangenome was defined as the difference between the core pangenome and the total number of genes in the pangenome. The core ratio was the defined as the number of genes in the ISS flight core pangenome versus to all-strains (ISS and terrestrial) core pangenome. The core pangenome fraction was defined as the number of genes in the core pangenome divided by the total pangenome size (core plus accessory).

For the functional enrichment analysis, the full bacterial genome with prophage regions was used for analysis. Functional enrichments between flight and terrestrial strains were determined using the built in anvi-compute-functional-enrichments-in-pan workflow[59] with COG20_FUNCTION as the annotation source. This workflow used the two-tailed Rao test statistic to determine the significance of a categorical variable (in this case flight versus terrestrial) to explain the functional occurrence. The resulting $p$ value was corrected for multiple tests and reported as a $q$ value, with a set significance threshold of $q < 0.05$. For species with subsampled genomes, a COG was considered enriched if it was present at a $q$ value ($p$ value adjusted for multiple tests) of 0.05 or lower for at least two subsets. Enrichment summaries and outputs from the anvio-summarize script were used for downstream analysis.

### Mitomycin C and UV treatment of ISS bacterial isolates
Flight isolates of *Paenibacillus polymyxa* (strains IIFSW-B4 and IIF8SW-P3), *Bacillus amyloliquefaciens* (strains IIF7SW-P4 and IIF7SC-B1), *Pseudomonas fulva* strain F8_7S_9B, and *Staphylococcus saprophyticus* (strains F5_7S_P13 and F6_7S_P5) were provided by the Biotechnology and Planetary Protection Group at the NASA Jet Propulsion Laboratory and grown overnight in tryptic soy broth (TSB) at 30 °C. Cultures were diluted to an OD600 of 0.3–0.4 and 4 mL was placed in a 30 mm diameter plastic petri dish. With the lid removed, the cultures were

subjected to aggregate UV doses of 5, 25, 50, 100, and 200 J m$^{-2}$ of UV-C radiation from a G30T8 UV lamp (75 μW cm$^{-2}$ at the plate surface). Doses were determined using a 2π UV-B/C light probe connected to a hand-held light meter (ILT2400, International Light Technologies, Peabody, MA, USA) in integrate mode. After each dose was delivered, duplicate 200 μL aliquots were removed from the petri dish and placed into separate wells of a 96-well microtiter plate. Cell growth was determined by microplate reader optical measurements at 600 nm once per hour (SpectraMax 340, Molecular Devices, San Jose, CA, USA). Negative control sample OD600 was subtracted from the time-course culture data prior to analysis.

For the mitomycin C treatment, flight isolates grown overnight in TSB at 30 °C were diluted to 0.3–0.4 OD600 and 200 μL was aliquoted into a 96-well plate. Mitomycin C stock solution in DMSO (10 mg/mL, Sigma-Aldrich, St. Louis, MO, USA) was diluted to 110 μg/mL in TSB before being serial diluted five times at 5× dilution in TSB. 5 μL of these dilutions, along with a TSB control, were added to separate wells of the culture-containing 96-well plate (2 technical replicates per isolate) resulting in a final concentration set of 500, 100, 20, 4, 0.8, and 0.0 ng mL$^{-1}$. Cell growth was determined by microplate reader optical measurements at 600 nm once per hour (SpectraMax 340, Molecular Devices, San Jose, CA, USA). Negative control sample OD600 was subtracted from the time-course culture data prior to analysis.

For phage isolation attempts, 5 mL of cultures were grown to mid-log phase before being treated with mitomycin C (0.5 μg/mL final concentration). Cultures were returned to the 30 °C incubator overnight. Cells and debris were removed by centrifugation and the supernatant was filtered through a low-protein binding 0.22 μm PES syringe filter. To the clarified supernatant, a PEG8000, NaCl stock solution was added to yield a final concentration of 8% PEG8000, 1 M NaCl. Solutions were incubated overnight at 4 °C, transferred to triplicate 1.5 mL microcentrifuge tubes and centrifuged at 15 000 × $g$ for 30 min at 4 °C. Tubes were visually examined for the presence of phage pellets.

### Long-read genome sequencing of bacterial isolates

Flight isolates of *Paenibacillus polymyxa* (strains IIFSW-B4 and IIF8SW-P3), *Bacillus amyloliquefaciens* (strains IIF7SW-P4 and IIF7SC-B1), *Pseudomonas fulva* strain F8_7S_9B, and *Staphylococcus saprophyticus* (strains F5_7S_P13 and F6_7S_P5) were grown on tryptic soy agar plates. After 72 h the plates were scraped and biomass was suspended in 1 mL of buffer (100 mM Tris HCl, 50 mM NaCl, pH 7.0), and genomic DNA was extracted via a lysozyme, proteinase K treatment followed by phenol-chloroform extraction and ethanol precipitation as outlined in[61]. DNA quality was assessed by fluorometric quantification (Qubit, Invitrogen, Waltham, MA, USA) and UV-Vis microvolume spectrophotometry (Nanodrop, Thermo Fischer Scientific, Waltham, MA, USA). *Paenibacillus polymyxa* and *Bacillus amyloliquefaciens*, as well as *Staphylococcus saprophyticus* F5_7S_P13, were excluded from the sequencing run due to biofilm polysaccharides preventing extraction of gDNA with sufficient purity for MinION sequencing. All gDNA manipulations were performed using wide-bore pipet tips to prevent genomic DNA (gDNA) shearing. Native gDNA was sequenced on a MinION R9.4 flowcell (Oxford Nanopore Technologies (ONT), Oxford, UK). The sequencing library was prepared using the ONT Rapid Barcoding Sequencing kit (SQK-RBK004, Oxford Nanopore Technologies, Oxford, UK) according to the manufacturer's protocol. Sequencing Fast5 files were base-called using the ONT Guppy base-caller (v6.2.1) using the super high accuracy model. Genomes were assembled using Flye [v2.9.1[62]] and polished with the ONT tool Medaka (v1.7.0, https://github.com/nanoporetech/medaka) using default parameters for both. Closed contigs were checked for prophage regions using the PHASTER web-based interface as outlined above.

### Functional annotation of prophage regions

Open reading frames encoded within the prophage nucleotide sequences from the PHASTER output were determined using the NCBI ORFinder[18]. Open reading frames were annotated using Microbe-Annotator [2.0.5,[63]], using DIAMOND search for homology determination[58]. The output file was combined with the $E$ values for the homology scores and any $E$ value higher than $1 \times 10^{-40}$ was flagged for manual curation. Proteins flagged for manual curation were compiled into a single file in fasta format and searched against the NCBI RefSeq database. The results were curated for accuracy and confidence and the annotation added to the summary file. All annotations in the summary file were then manually assigned to one of 21 categories based on their functional annotation.

All data visualization was performed using in Python (v3.9) using Jupyter Notebooks [v1.0.0,[64]] and the packages Matplotlib [v3.5.1,[65]] and Seaborn [v0.11.2,[66]]. Additional packages used for analysis included NumPy [v1.22.3,[67]], Pandas [v1.4.2,[68,69]], and Biopython [1.79,[70]]. No custom code, algorithms, or scripts central to the research were generated.

### Reporting summary

Further information on research design is available in the Nature Portfolio Reporting Summary linked to this article.

## Data availability

The long-read assemblies and raw fast5 files generated in this study have been deposited in the NASA Open Science Data Repository (https://osdr.nasa.gov/) under accession code OSD-582 (https://osdr.nasa.gov/bio/repo/data/studies/OSD-582, https://doi.org/10.26030/v6s6-w170) and in the NCBI databases Assembly under accession codes GCF_017167745.1 (https://www.ncbi.nlm.nih.gov/datasets/genome/GCF_017167745.1) and GCF_017166245.1 (https://www.ncbi.nlm.nih.gov/datasets/genome/GCF_017166245.1), BioProject under accession code PRJNA994947 (https://www.ncbi.nlm.nih.gov/bioproject/?term=PRJNA994947), BioSamples under accession codes SAMN36453224 (https://www.ncbi.nlm.nih.gov/biosample/36453225) and SAMN36453224 (https://www.ncbi.nlm.nih.gov/biosample/36453224), and SRA under accession codes SRR25281509 (https://www.ncbi.nlm.nih.gov/sra/SRX21026417[accn]) and SRR25281510 (https://www.ncbi.nlm.nih.gov/sra/SRX21026416[accn]). All data for figures have been uploaded to Figshare (https://doi.org/10.6084/m9.figshare.23290226). Source data are provided with this paper.

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

## Acknowledgements

The authors would like to acknowledge Dr. Kasthuri Venkateswaran and Dr. Anna Simpson at the NASA Jet Propulsion Laboratory for providing the ISS flight isolates and Dr. Braden Tierney at Weill-Cornell Medical College for his critical review of the paper. This work was supported by the NASA Space Biology Space Life Science Training Program and the NASA Human Research Program Model Translation and Space Biology Integration project.

## Author contributions

J.T.B. and I.I. conceived and designed the study. J.T.B. performed the pangenome analysis, mitomycin C and UV experiments, MinION sequencing and assembly, and curation of prophage-encoded protein functions. I.I. built and executed the pipeline for prophage identification, comparison, and classification, and the presence of antibiotic resistance markers. Both authors wrote and edited the paper.

## Competing interests

The authors declare no competing interests.
