## [Peer Review File · Nature Communications]

REVIEWER COMMENTS

Reviewer #1 (Remarks to the Author):

The manuscript describes the identification and characterization of putative prophages found in different bacterial species isolated on the ISS. The authors present data on the identification, annotation and comparison of prophage sequences. They list a number of characteristics that are unique to some of the prophages, including potential AMR and virulence-associated genes. The authors suggest that these unique genetic features carried by prophages that are found only in spaceflight bacterial strains could be linked somehow to space adaptation.

The topic is original and very interesting, and the authors have done a good amount of work to identify and annotate prophage genomes and try to identify features that could be of interest for future characterization studies.

The manuscript is generally well presented and written. However, there are a number of elements that need to be revised/corrected and/or improved.

General comments:

One important weakness in the methodology is that there is no indication about the genetic relatedness of the strains selected for the study. For example, how genetically related are the spaceflight strains compared to the terrestrial strains in terms of core genome? If this information is already available from previous papers, the authors should provide this information with proper reference. For example, in Fig.2, 20/21 strains of *A. pittii* contain 6 prophages and only one strain contains 7. Are the 20 strains highly similar? In fact, supplementary table 1 clearly shows that many of the isolates for a given species were isolated the same day from the same location. Hence, these strains are likely very similar or even possibly clones of the same strain. Terrestrial isolates are probably more diverse and as such, more prophage diversity was observed. Interestingly, the number of prophages/genome in *S. epidermidis* and *S. saprophyticus* from the ISS is higher and more diverse and might simply be the result of a better sampling campaign, especially *S. epidermidis* for which several isolates were collected over several years and various locations. A bacterial core genome phylogenetic analysis, excluding prophages, would be highly relevant to show how terrestrial and ISS bacterial genomes are related to each other. It might even show whether some prophages can infect related but different strains within the ISS, suggesting active horizontal transfer. If the ISS bacterial genomes are less diversified and are highly clonal, it is normal that the number and diversity of detected prophages is similar (fig. 2).

Another major weakness is the methodology for prophage induction and phage detection. Not enough technical details are presented, and it is difficult to assess whether the identified prophages are functional or not. The authors have tried UV induction but how the presence of phage plaques was tested is not well explained. This will need to be clarified and improved.

The authors suggest that prophage-encoded functions could provide some benefits to the bacterial host. This is generally well-recognized in the literature, but although the authors acknowledge in the discussion that further experimentation will be required, the bioinformatic analyses are highly speculative and often rely only on blast results, and data do not always support the interpretations. For example, the genes identified as involved in AMR are not well-described and their function is based solely on blast on Refseq proteins. The use of more specialized databases like CARD could be more informative. Also, providing some information about how these genes could participate to AMR, or against which antibiotics these genes would confer resistance would be required.

Other specific comments:

L138: Taxonomy has changed in 2021. Please revise and correct taxonomy according to the new ICTV classification (orders and families don't exist anymore, so Caudovirales does not exist anymore, as well as *Siphoviridae* and *Myoviridae* families.).

L140: "... 71 phage clusters"?

L141: "... of 95% nucleotide identity for..."

L142: What is meant by "if they became lytic"?

L149: "...the 95% nucleotide identity cutoff..."

L156: Which strains (ID number) were tested for prophage induction and for phage plaquing? The authors have tested 7 strains from 4 different species, meaning that some species had only one representative strain. Normally, when looking for inducible prophages, a lysogenic host is induced by UV, mitomycin C or other stresses, and the released phages are plated onto a different strain of the same species to detect plaque formation. Otherwise, the original lysogenic strain is naturally resistant to reinfection by the same phage. There is not enough methodological detail provided in this part. Plaques don't form on colonies; they form on bacterial lawns. It is not clear from the materials and methods and

the results section what the authors want to explain and whether a soft agar overlay method was used or not. Also, UV is one method to induce prophages, but other common methods like mitomycin C or other DNA-damaging antibiotics are also frequently used. At least mitomycin C should have been tested before concluding on the functionality of the identified phages.

L160: Using broth cultures, it is possible to observe typical bacterial lysis (sharp drop in optical density) caused by prophage induction, when the appropriate inducer is used, and when induction occurs. Are the methods for prophage induction and phage plaque detection already known for the selected bacterial species? Were appropriate conditions used to detect phages? More details are required to assess whether the experiments were conducted adequately or not.

L161-162: How MinION sequencing can inform on inducible prophages? The justification for MinION sequencing is totally unrelated to prophage induction. MinION sequencing did not further demonstrate the inducibility of the prophages. This part should be rewritten to reflect that the authors performed MinION for other reasons than because phage plaques were not detected after prophage induction.

L169: how the authors explain that the short-read incomplete prophage assembly is missing from the MinION long-read genome?

L171: If non-closed genomes were used to search for prophages, it is not surprising that some prophages may have been missed because they were split across two or more contigs. PHASTER considers a number of criteria and below a certain score, it will not retrieve significant results. It seems from the materials and methods section that “complete genomes” (line 289) were used, but the results section suggests the contrary. This should be better explained in the materials and methods and in the results section, i.e. whether complete and closed genomes or scaffolds were used for PHASTER analysis.

L172: “... the incomplete prophage #5 from ...”

L176: Is the short-read incomplete prophage missing or it simply now maps onto another prophage region in the long-read assembly?

L181-182: Shouldn't we read “... as a single 47.6 kb region in the long-read assembly but as two prophage regions in the short-read assembly...”?

L211: Searching for AMR genes and virulence factors should be done using tools such as ABRICATE that combine multiple databases. Simple blast on NCBI can lead to false calls. It is possible that genes annotated as AMR are not real AMR genes when analyzed using AMR-specific databases with curated genes.

L214: "... genes part of pathogenicity islands and..."

L251: A more thorough screening of AMR and virulence factor should be performed using curated databases.

L252: "In comparison, ... terrestrial ..." something is missing in the sentence.

L257: There is no experimental evidence that the identified genes are indeed involved in AMR. Based on the annotation, it is not obvious to which antibiotic these putative AMR genes would provide resistance to. Providing this information would be necessary.

L270-272: There is no experimental evidence to support this hypothesis. Other methods of phage detection should be used. For example, the prophage sequences are known, so the authors could induce their lysogenic strains and try to detect by PCR the induced phages in the lysate, after adequate DNase/RNase treatment and proper controls to detect bacterial DNA contamination. Whole phage DNA/RNA extraction can even be done on induction lysates and restriction profiles can be done. The authors cannot conclude that the prophages did not induce upon UV treatment without further experimentation and additional methodological details.

L289: by "complete genome" do you mean closed genome without gaps? If so, why some prophages were found to be split on two or more contigs after comparing with MinION sequencing?

L294: Were the results from PhiSpy equivalent to those from PHASTER? Were any discrepancies detected? This should be mentioned in the text. Also: "...confer fitness to various antibiotics". Verify sentence: ... confer resistance or tolerance to various antibiotics maybe?

L300: 90% nucleotide identity

L326-327: Additional experimental details are needed for the phage detection section.

L329: "... by phenol-chloroform..." ?

L332: All gDNA manipulations were performed.

All bacterial names should be in italic in the reference section.

Reviewer #2 (Remarks to the Author):

The manuscript by Irby and Broddrick sequenced 245 genomes and obtained the metagenome-assembled genomes from bacteria collected in the International Space Station. The authors found the lysogeny prophages are different in spaceflight strains and their terrestrial counterparts. Overall, the study is interesting and the writing is very clear. However, the manuscript has two major concerns: one that the major conclusion is based on the comparison of the sequenced spaceflight strains and the sequenced terrestrial strains. Differences of prophages are common in microbiomes isolated from two different environmental niches on earth, thus this conclusion seems premature without proper control strains. The second is the Program PHASTER and RefSeq Genome Database used to predict prophages have limitations. It is important have to a comprehensive or a combination of different programs to get a more accurate analysis of complete and incomplete prophage in these strains and for the prediction of prophage encoded functional genes.

REVIEWER COMMENTS

Reviewer #1 (Remarks to the Author):

The manuscript describes the identification and characterization of putative prophages found in different bacterial species isolated on the ISS. The authors present data on the identification, annotation and comparison of prophage sequences. They list a number of characteristics that are unique to some of the prophages, including potential AMR and virulence-associated genes. The authors suggest that these unique genetic features carried by prophages that are found only in spaceflight bacterial strains could be linked somehow to space adaptation.

The topic is original and very interesting, and the authors have done a good amount of work to identify and annotate prophage genomes and try to identify features that could be of interest for future characterization studies.

The manuscript is generally well presented and written. However, there are a number of elements that need to be revised/corrected and/or improved.

General comments:

One important weakness in the methodology is that there is no indication about the genetic relatedness of the strains selected for the study. For example, how genetically related are the spaceflight strains compared to the terrestrial strains in terms of core genome? If this information is already available from previous papers, the authors should provide this information with proper reference. For example, in Fig.2, 20/21 strains of *A. pittii* contain 6 prophages and only one strain contains 7. Are the 20 strains highly similar? In fact, supplementary table 1 clearly shows that many of the isolates for a given species were isolated the same day from the same location. Hence, these strains are likely very similar or even possibly clones of the same strain. Terrestrial isolates are probably more diverse and as such, more prophage diversity was observed. Interestingly, the number of prophages/genome in *S. epidermidis* and *S. saprophyticus* from the ISS is higher and more diverse and might simply be the result of a better sampling campaign, especially *S. epidermidis* for which several isolates were collected over several years and various locations. A bacterial core genome phylogenetic analysis, excluding prophages, would be highly relevant to show how terrestrial and ISS bacterial genomes are related to each other. It might even show whether some prophages can infect related but different strains within the ISS, suggesting active horizontal transfer. If the ISS bacterial genomes are less diversified and are highly clonal, it is normal that the number and diversity of detected prophages is similar (fig. 2).

We agree with the reviewer that the genomic proximity of the flight strains provide important context for our analysis. In fact, we have performed this analysis, using average nucleotide identity. We did not include it in the initial submission due to manuscript length and figure requirements (the original submission was at a different Nature journal). However, with the importance of the analysis and the increased article length afforded us by Nature Communications, we have added this analysis to the manuscript, using a pangenome approach without the prophage regions as suggested.

Another major weakness is the methodology for prophage induction and phage detection. Not enough technical details are presented, and it is difficult to assess whether the identified prophages are

functional or not. The authors have tried UV induction but how the presence of phage plaques was tested is not well explained. This will need to be clarified and improved.

We understand the reviewer's confusion regarding this section. We have edited the text appropriately to include the rationale and methods for phage induction. Additionally, we performed mitomycin C treatment of the flight strains, along with attempts to isolate phage via filtration and PEG precipitation.

The authors suggest that prophage-encoded functions could provide some benefits to the bacterial host. This is generally well-recognized in the literature, but although the authors acknowledge in the discussion that further experimentation will be required, the bioinformatic analyses are highly speculative and often rely only on blast results, and data do not always support the interpretations. For example, the genes identified as involved in AMR are not well-described and their function is based solely on blast on Refseq proteins. The use of more specialized databases like CARD could be more informative. Also, providing some information about how these genes could participate to AMR, or against which antibiotics these genes would confer resistance would be required.

We agree that the AMR and virulence genes discussed are putative and, based on RefSeq annotation, few would be considered functional antibiotic resistance genes. Those that we identified as resistance genes are the drug efflux pumps identified in *B. amyloliquifaciens*, *B. thuringiensis*, *K. quasipneumoniae*, and *S. epidermidis* as well as the two antibiotic resistance genes found, an aminoglycoside resistance gene in *P. polymyxa* and a trimethoprim resistance gene in *S. epidermidis*. These changes are now reflected in the supplemental material to show the distinction between antibiotic resistance genes and virulence genes.

AMR gene presence results were corroborated using ABRicate, a package which uses various antibiotic resistance databases including NCBI, CARD, ARG-ANNOT, Resfinder, MEGARES, EcoOH, PlasmidFinder, Ecoli_VF and VFDB. Through ABRicate, we were able to confirm the presence of the aminoglycoside resistance gene in *P. polymyxa*, a macrolide resistance gene rather than a trimethoprim resistance gene in *S. epidermidis*, and an efflux pump in *S. epidermidis*. We have edited the text as appropriate to include this additional analysis.

Other specific comments:

L138: Taxonomy has changed in 2021. Please revise and correct taxonomy according to the new ICTV classification (orders and families don't exist anymore, so Caudovirales does not exist anymore, as well as *Siphoviridae* and *Myoviridae* families.).

Changed the order and families to the correct class *Caudoviricetes*. These changes are also reflected in the supplemental data.

L140: "... 71 phage clusters"?

We have re-written this as "intact prophage clusters" to hopefully increase the clarity of the text.

L141: "... of 95% nucleotide identity for..."

We have re-written this as "95% query identity (nucleotide identity multiplied by alignment percentage)" to hopefully increase the clarity of the text.

L142: What is meant by “if they became lytic”?

We were attempting to illustrate that this comparison is based on prophage regions and not isolated viral particles. However, the nuance causes confusion so we have removed this portion of the sentence.

L149: “...the 95% nucleotide identity cutoff...”

We have re-written this as “95% query identity cutoff” to hopefully increase the clarity of the text.

L156: Which strains (ID number) were tested for prophage induction and for phage plaquing? The authors have tested 7 strains from 4 different species, meaning that some species had only one representative strain. Normally, when looking for inducible prophages, a lysogenic host is induced by UV, mitomycin C or other stresses, and the released phages are plated onto a different strain of the same species to detect plaque formation. Otherwise, the original lysogenic strain is naturally resistant to reinfection by the same phage. There is not enough methodological detail provided in this part. Plaques don't form on colonies; they form on bacterial lawns. It is not clear from the materials and methods and the results section what the authors want to explain and whether a soft agar overlay method was used or not. Also, UV is one method to induce prophages, but other common methods like mitomycin C or other DNA-damaging antibiotics are also frequently used. At least mitomycin C should have been tested before concluding on the functionality of the identified phages.

We thank the reviewer for the thoughtful suggestions and insights on this section. Indeed, it was poorly communicated and incomplete. We have added sections on mitomycin C and UV treatment of the cultures and included more complete methodological details. The reviewer is correct that for the species for which we had two isolates, the strains we received from JPL contained the same complete prophages and thus were resistant to cross infection via a soft agar method. We instead used the OD-based mechanism, suggested below and included results, supplemental figures, and methods consistent with these suggestions.

L160: Using broth cultures, it is possible to observe typical bacterial lysis (sharp drop in optical density) caused by prophage induction, when the appropriate inducer is used, and when induction occurs. Are the methods for prophage induction and phage plaque detection already known for the selected bacterial species? Were appropriate conditions used to detect phages? More details are required to assess whether the experiments were conducted adequately or not.

The methods for prophage induction are unknown in these species. We explored UV and mitomycin C treatment with OD-based assessments of bacterial lysis, consistent with the above suggestion.

L161-162: How MinION sequencing can inform on inducible prophages? The justification for MinION sequencing is totally unrelated to prophage induction. MinION sequencing did not further demonstrate the inducibility of the prophages. This part should be rewritten to reflect that the authors performed MinION for other reasons than because phage plaques were not detected after prophage induction. We have rewritten this section as suggested. In the past, we have sequenced bacteriophage in environmental bacterial metagenomes with the MinION. We were hoping that UV induction would have produced sufficient viral particles that they would have been detected along with the bacterial DNA. However, this didn't occur and with the suggestions regarding OD-based methods we have rewritten the manuscript as suggested.

L169: how the authors explain that the short-read incomplete prophage assembly is missing from the MinION long-read genome?

We've expanded this section to include discussion that the entire prophage region mapped to the long read assembly, but this particular assembly was low coverage (18x) and the mapped prophage region had the highest number of mismatches and gaps of those analyzed. Still, PhiSpy did annotate this region as a prophage, so the assembly accuracy isn't a completely satisfying explanation. At the same time, an analysis as to why one algorithm failed to annotate the region while the other didn't is out of scope. Reassuringly, the PHASTER called intact prophages were all consistent with PhiSpy, which we mention in the text.

L171: If non-closed genomes were used to search for prophages, it is not surprising that some prophages may have been missed because they were split across two or more contigs. PHASTER considers a number of criteria and below a certain score, it will not retrieve significant results. It seems from the materials and methods section that "complete genomes" (line 289) were used, but the results section suggests the contrary. This should be better explained in the materials and methods and in the results section, i.e. whether complete and closed genomes or scaffolds were used for PHASTER analysis. We understand the reviewer's confusion. The ISS flight genomes have an assembly level no better than "contig" or "scaffold" while the terrestrial genomes used for prophage comparison were all "complete." We have edited the text to show the assembly levels for different analysis as it changes depending on the analysis and the flight versus terrestrial genomes.

L172: "... the incomplete prophage #5 from ..."

We appreciate the suggested edit and have included it in the text.

L176: Is the short-read incomplete prophage missing or it simply now maps onto another prophage region in the long-read assembly?

Similar to the example in *S. saprophyticus*, this prophage was missing in the long read assembly with no clear reason for its exclusion from the prophage calls. We have indicated such in the manuscript.

L181-182: Shouldn't we read "... as a single 47.6 kb region in the long-read assembly but as two prophage regions in the short-read assembly...?"

Yes, we thank the reviewer for catching this oversight.

L211: Searching for AMR genes and virulence factors should be done using tools such as ABRICATE that combine multiple databases. Simple blast on NCBI can lead to false calls. It is possible that genes annotated as AMR are not real AMR genes when analyzed using AMR-specific databases with curated genes.

We agree with this suggestion and edited the text to reflect the AMR and virulence genes identified using ABRicate.

L214: "... genes part of pathogenicity islands and..."

Edit accepted and changed.

L251: A more thorough screening of AMR and virulence factor should be performed using curated databases.

We agree with the above statement, and the genes are now confirmed using ABRicate and the manuscript updated accordingly.

L252: "In comparison, ... terrestrial ..." something is missing in the sentence. The word "terrestrial" was a remnant of an edit. It has been removed.

L257: There is no experimental evidence that the identified genes are indeed involved in AMR. Based on the annotation, it is not obvious to which antibiotic these putative AMR genes would provide resistance to. Providing this information would be necessary.

We agree with this critique. We have changed the wording to reflect the uncertain nature of the resistance genes and discuss the high confidence AMR annotations to include the antibiotic resistance associated with those genes.

L270-272: There is no experimental evidence to support this hypothesis. Other methods of phage detection should be used. For example, the prophage sequences are known, so the authors could induce their lysogenic strains and try to detect by PCR the induced phages in the lysate, after adequate DNase/RNase treatment and proper controls to detect bacterial DNA contamination. Whole phage DNA/RNA extraction can even be done on induction lysates and restriction profiles can be done. The authors cannot conclude that the prophages did not induce upon UV treatment without further experimentation and additional methodological details.

We agree this comment was speculative and have stricken it from the manuscript.

L289: by "complete genome" do you mean closed genome without gaps? If so, why some prophages were found to be split on two or more contigs after comparing with MinION sequencing?

We have edited the text for clarity. The ISS flight genomes are all "scaffold" or "contig" so while we were able to use "complete" genomes from the terrestrial dataset, the flight genomes were all incomplete. This motivated the use of long-read assemblies, which is also now in the text.

L294: Were the results from PhiSpy equivalent to those from PHASTER? Were any discrepancies detected? This should be mentioned in the text. Also: "...confer fitness to various antibiotics". Verify sentence: ... confer resistance or tolerance to various antibiotics maybe?

We have updated the text to indicate where PhiSpy and PHASTER differed and the rationale for using each. In particular, we were encouraged that all the PHASTER "intact" prophages were consistent with PhiSpy. We also edited the sentence to read "associated with putative virulence or antibiotic resistance functions"

L300: 90% nucleotide identity
Edit accepted and changed.

L326-327: Additional experimental details are needed for the phage detection section.
This section has been updated with the details on the mitomycin C and UV experiments.

L329: "... by phenol-chloroform..." ?
Edit accepted and changed.

L332: All gDNA manipulations were performed.
Edit accepted and changed.

All bacterial names should be in italic in the reference section.
Edit accepted and changed.

Reviewer #2 (Remarks to the Author):

The manuscript by Irby and Broddrick sequenced 245 genomes and obtained the metagenome-assembled genomes from bacteria collected in the International Space Station. The authors found the lysogeny prophages are different in spaceflight strains and their terrestrial counterparts. Overall, the study is interesting and the writing is very clear. However, the manuscript has two major concerns: one that the major conclusion is based on the comparison of the sequenced spaceflight strains and the sequenced terrestrial strains. Differences of prophages are common in microbiomes isolated from two different environmental niches on earth, thus this conclusion seems premature without proper control strains. The second is the Program PHASTER and RefSeq Genome Database used to predict prophages have limitations. It is important have to a comprehensive or a combination of different programs to get a more accurate analysis of complete and incomplete prophage in these strains and for the prediction of prophage encoded functional genes.

Regarding the computational pipeline, we have expanded our workflow to include PhiSpy in addition to PHASTER and limiting our intact/complete prophage analysis to those that were consistent with both tools. Reassuringly, all our previously identified intact prophage regions were consistent across both algorithms. We also added AMRFinder to our pipeline, which includes genes from the Comprehensive Antibiotic Resistance Database (CARD), confirmed literature review by database curators, and collaboration with Gen-FS and subject matter experts.

Additionally, we agree that environmental differences in phage is common; however, our primary focus was on the novelty of the prophages in ISS isolates. Our conclusion that the functional content of those prophage regions is correlated to fitness in the built spacecraft environment reflects our agreement that additional work is needed to make more mature claims.

REVIEWER COMMENTS

Reviewer #3 (Remarks to the Author):

As suggested by the editor, I have read this revised manuscript and primarily looked at how Reviewer 1's comments have been addressed. Generally speaking, the authors have addressed the majority of issues raised and have done a commendable job with the revisions.

That being said, there are still three outstanding issues in my opinion:

1. There is insufficient detail provided about the terrestrial genomes used in this study. Rather than using all the genomes from Refseq, it may be more advisable to use a diverse set of genomes from NCBI Genomes or even expand to IMG/M. Additionally, I would suggest the authors to include all the genomes names with accession numbers in Supplementary Table 1. I think this would go above and beyond in demonstrating the differences between the ISS and terrestrial strains.
2. As of now, only a part of the data is only available on NASA's data repository. I think this is not sufficient. First, the prophage sequences must be submitted to NCBI and made public. Additionally, for the purpose of ease of use and reproducibility, I would encourage the authors to submit the minion assemblies, raw data, and genomes to NCBI as well. At the end of the day, this will only benefit the authors through increased visibility on NCBI searches and higher citations.
3. In my opinion, the IMG VR database is the current gold standard for viral genomics. I would suggest that the authors include an analysis of prophage homology with this database in addition the one presented with NCBI Refseq.

REVIEWER COMMENTS ROUND 2

Reviewer #3 (Remarks to the Author):

As suggested by the editor, I have read this revised manuscript and primarily looked at how Reviewer 1's comments have been addressed. Generally speaking, the authors have addressed the majority of issues raised and have done a commendable job with the revisions.

That being said, there are still three outstanding issues in my opinion:

1. There is insufficient detail provided about the terrestrial genomes used in this study. Rather than using all the genomes from Refseq, it may be more advisable to use a diverse set of genomes from NCBI Genomes or even expand to IMG/M. Additionally, I would suggest the authors to include all the genomes names with accession numbers in Supplementary Table 1. I think this would go above and beyond in demonstrating the differences between the ISS and terrestrial strains.

This comment was indeed an oversight on our part, we have included the accession and genome names of the terrestrial genomes in the Supplemental Table. We compared the Refseq genomes to those in IMG/M and only 71 genomes were unique to IMG/M, compared to RefSeq, for our chosen species (<5%). While including these genomes would increase the diversity of the terrestrial genome comparison, we did not have sufficient resources to re-run our pipeline with these genomes included.

Our rationale for using the Refseq database, is that completeness and contiguity of the genomes affects our analysis. While expanding to more genomes in NCBI increases the diversity, the gaps in those genomes, for example from metagenomes, creates noise in the pangenome analysis. We don't know if the difference between flight and terrestrial is due to actual differences in genome content or due to assembly gaps. We have included a comment in the Discussion regarding the limitations of selecting a single database.

2. As of now, only a part of the data is only available on NASA's data repository. I think this is not sufficient. First, the prophage sequences must be submitted to NCBI and made public. Additionally, for the purpose of ease of use and reproducibility, I would encourage the authors to submit the minion assemblies, raw data, and genomes to NCBI as well. At the end of the day, this will only benefit the authors through increased visibility on NCBI searches and higher citations.

We have submitted the data and genomes to NCBI under the identifiers: BioProject PRJNA994947, BioSamples SAMN36453224 and SAMN36453224, and SRA Accessions SRR25281509 and SRR25281510. The phage sequences were not sequenced from a viral particle. As such, we instead felt it was more appropriate to include the genome region for each prophage in the supplemental table. This way the regions can be found on existing NCBI assemblies, but we don't introduce confusion by creating additional sequence accessions. Our on-going work to isolate and sequence intact phage particles, if successful, will certainly be uploaded as new sequences to relevant databases.

Additionally all data for the Figures have been uploaded to Figshare. The DOI for the Figshare data and NCBI identifiers have been added to the Data Availability Statement in the manuscript.

3. In my opinion, the IMG VR database is the current gold standard for viral genomics. I would suggest that the authors include an analysis of prophage homology with this database in addition the one presented with NCBI Refseq.

We have performed this analysis and added the homology of the flight prophages to the IMG VR DB as a column in Table 1. The IMG VR database was indeed more comprehensive, and we have shown this comparison in the manuscript. As a result of this analysis, the data in Table 1 has changed as has the data for Figure 4. Overall, the conclusion that flight bacterial prophages are novel is maintained; however, due to the IMG VR database we were able to ascribe a baseline taxonomy to more of the prophage regions.

Previous Revision Responses

Reviewer #1 (Remarks to the Author):

The manuscript describes the identification and characterization of putative prophages found in different bacterial species isolated on the ISS. The authors present data on the identification, annotation and comparison of prophage sequences. They list a number of characteristics that are unique to some of the prophages, including potential AMR and virulence-associated genes. The authors suggest that these unique genetic features carried by prophages that are found only in spaceflight bacterial strains could be linked somehow to space adaptation.

The topic is original and very interesting, and the authors have done a good amount of work to identify and annotate prophage genomes and try to identify features that could be of interest for future characterization studies.

The manuscript is generally well presented and written. However, there are a number of elements that need to be revised/corrected and/or improved.

General comments:

One important weakness in the methodology is that there is no indication about the genetic relatedness of the strains selected for the study. For example, how genetically related are the spaceflight strains compared to the terrestrial strains in terms of core genome? If this information is already available from previous papers, the authors should provide this information with proper reference. For example, in Fig.2, 20/21 strains of *A. pittii* contain 6 prophages and only one strain contains 7. Are the 20 strains highly similar? In fact, supplementary table 1 clearly shows that many of the isolates for a given species were isolated the same day from the same location. Hence, these strains are likely very similar or even possibly clones of the same strain. Terrestrial isolates are probably more diverse and as such, more prophage diversity was observed. Interestingly, the number of prophages/genome in *S. epidermidis* and *S. saprophyticus* from the ISS is higher and more diverse and might simply be the result of a better sampling campaign, especially *S. epidermidis* for which several isolates were collected over several years and various locations. A bacterial core genome phylogenetic analysis, excluding prophages, would be highly relevant to show how terrestrial and ISS bacterial genomes are related to each other. It might even show whether some prophages can infect related but different strains within the ISS, suggesting active horizontal transfer. If the ISS bacterial genomes are less diversified and are highly clonal, it is normal that the number and diversity of detected prophages is similar (fig. 2).

We agree with the reviewer that the genomic proximity of the flight strains provide important context for our analysis. In fact, we have performed this analysis, using average nucleotide identity. We did not include it in the initial submission due to manuscript length and figure requirements (the original submission was at a different Nature journal). However, with the importance of the analysis and the increased article length afforded us by Nature Communications, we have added this analysis to the manuscript, using a pangenome approach without the prophage regions as suggested.

Another major weakness is the methodology for prophage induction and phage detection. Not enough technical details are presented, and it is difficult to assess whether the identified prophages are functional or not. The authors have tried UV induction but how the presence of phage plaques was tested is not well explained. This will need to be clarified and improved.

We understand the reviewer's confusion regarding this section. We have edited the text appropriately to include the rationale and methods for phage induction. Additionally, we performed mitomycin C treatment of the flight strains, along with attempts to isolate phage via filtration and PEG precipitation.

The authors suggest that prophage-encoded functions could provide some benefits to the bacterial host. This is generally well-recognized in the literature, but although the authors acknowledge in the discussion that further experimentation will be required, the bioinformatic analyses are highly speculative and often rely only on blast results, and data do not always support the interpretations. For example, the genes identified as involved in AMR are not well-described and their function is based solely on blast on Refseq proteins. The use of more specialized databases like CARD could be more informative. Also, providing some information about how these genes could participate to AMR, or against which antibiotics these genes would confer resistance would be required.

We agree that the AMR and virulence genes discussed are putative and, based on RefSeq annotation, few would be considered functional antibiotic resistance genes. Those that we identified as resistance genes are the drug efflux pumps identified in *B. amyloliquifaciens*, *B. thuringiensis*, *K. quasipneumoniae*, and *S. epidermidis* as well as the two antibiotic resistance genes found, an aminoglycoside resistance gene in *P. polymyxa* and a trimethoprim resistance gene in *S. epidermidis*. These changes are now reflected in the supplemental material to show the distinction between antibiotic resistance genes and virulence genes.

AMR gene presence results were corroborated using ABRicate, a package which uses various antibiotic resistance databases including NCBI, CARD, ARG-ANNOT, Resfinder, MEGARES, EcoOH, PlasmidFinder, Ecoli_VF and VFDB. Through ABRicate, we were able to confirm the presence of the aminoglycoside resistance gene in *P. polymyxa*, a macrolide resistance gene rather than a trimethoprim resistance gene in *S. epidermidis*, and an efflux pump in *S. epidermidis*. We have edited the text as appropriate to include this additional analysis.

Other specific comments:

L138: Taxonomy has changed in 2021. Please revise and correct taxonomy according to the new ICTV classification (orders and families don't exist anymore, so Caudovirales does not exist anymore, as well as *Siphoviridae* and *Myoviridae* families.).

Changed the order and families to the correct class *Caudoviricetes*. These changes are also reflected in the supplemental data.

L140: "... 71 phage clusters"?

We have re-written this as "intact prophage clusters" to hopefully increase the clarity of the text.

L141: "... of 95% nucleotide identity for..."

We have re-written this as "95% query identity (nucleotide identity multiplied by alignment percentage)" to hopefully increase the clarity of the text.

L142: What is meant by "if they became lytic"?

We were attempting to illustrate that this comparison is based on prophage regions and not isolated viral particles. However, the nuance causes confusion so we have removed this portion of the sentence.

L149: "...the 95% nucleotide identity cutoff..."

We have re-written this as "95% query identity cutoff" to hopefully increase the clarity of the text.

L156: Which strains (ID number) were tested for prophage induction and for phage plaquing? The authors have tested 7 strains from 4 different species, meaning that some species had only one representative strain. Normally, when looking for inducible prophages, a lysogenic host is induced by UV, mitomycin C or other stresses, and the released phages are plated onto a different strain of the same species to detect plaque formation. Otherwise, the original lysogenic strain is naturally resistant to reinfection by the same phage. There is not enough methodological detail provided in this part. Plaques don't form on colonies; they form on bacterial lawns. It is not clear from the materials and methods and the results section what the authors want to explain and whether a soft agar overlay method was used or not. Also, UV is one method to induce prophages, but other common methods like mitomycin C or other DNA-damaging antibiotics are also frequently used. At least mitomycin C should have been tested before concluding on the functionality of the identified phages.

We thank the reviewer for the thoughtful suggestions and insights on this section. Indeed, it was poorly communicated and incomplete. We have added sections on mitomycin C and UV treatment of the cultures and included more complete methodological details. The reviewer is correct that for the species for which we had two isolates, the strains we received from JPL contained the same complete prophages and thus were resistant to cross infection via a soft agar method. We instead used the OD-based mechanism, suggested below and included results, supplemental figures, and methods consistent with these suggestions.

L160: Using broth cultures, it is possible to observe typical bacterial lysis (sharp drop in optical density) caused by prophage induction, when the appropriate inducer is used, and when induction occurs. Are the methods for prophage induction and phage plaque detection already known for the selected bacterial species? Were appropriate conditions used to detect phages? More details are required to assess whether the experiments were conducted adequately or not.

The methods for prophage induction are unknown in these species. We explored UV and mitomycin C treatment with OD-based assessments of bacterial lysis, consistent with the above suggestion.

L161-162: How MinION sequencing can inform on inducible prophages? The justification for MinION sequencing is totally unrelated to prophage induction. MinION sequencing did not further demonstrate the inducibility of the prophages. This part should be rewritten to reflect that the authors performed

MinION for other reasons than because phage plaques were not detected after prophage induction. We have rewritten this section as suggested. In the past, we have sequenced bacteriophage in environmental bacterial metagenomes with the MinION. We were hoping that UV induction would have produced sufficient viral particles that they would have been detected along with the bacterial DNA. However, this didn't occur and with the suggestions regarding OD-based methods we have rewritten the manuscript as suggested.

L169: how the authors explain that the short-read incomplete prophage assembly is missing from the MinION long-read genome?

We've expanded this section to include discussion that the entire prophage region mapped to the long read assembly, but this particular assembly was low coverage (18x) and the mapped prophage region had the highest number of mismatches and gaps of those analyzed. Still, PhiSpy did annotate this region as a prophage, so the assembly accuracy isn't a completely satisfying explanation. At the same time, an analysis as to why one algorithm failed to annotate the region while the other didn't is out of scope. Reassuringly, the PHASTER called intact prophages were all consistent with PhiSpy, which we mention in the text.

L171: If non-closed genomes were used to search for prophages, it is not surprising that some prophages may have been missed because they were split across two or more contigs. PHASTER considers a number of criteria and below a certain score, it will not retrieve significant results. It seems from the materials and methods section that "complete genomes" (line 289) were used, but the results section suggests the contrary. This should be better explained in the materials and methods and in the results section, i.e. whether complete and closed genomes or scaffolds were used for PHASTER analysis. We understand the reviewer's confusion. The ISS flight genomes have an assembly level no better than "contig" or "scaffold" while the terrestrial genomes used for prophage comparison were all "complete." We have edited the text to show the assembly levels for different analysis as it changes depending on the analysis and the flight versus terrestrial genomes.

L172: "... the incomplete prophage #5 from ..."

We appreciate the suggested edit and have included it in the text.

L176: Is the short-read incomplete prophage missing or it simply now maps onto another prophage region in the long-read assembly?

Similar to the example in *S. saprophyticus*, this prophage was missing in the long read assembly with no clear reason for its exclusion from the prophage calls. We have indicated such in the manuscript.

L181-182: Shouldn't we read "... as a single 47.6 kb region in the long-read assembly but as two prophage regions in the short-read assembly...?"

Yes, we thank the reviewer for catching this oversight.

L211: Searching for AMR genes and virulence factors should be done using tools such as ABRICATE that combine multiple databases. Simple blast on NCBI can lead to false calls. It is possible that genes annotated as AMR are not real AMR genes when analyzed using AMR-specific databases with curated

genes.

We agree with this suggestion and edited the text to reflect the AMR and virulence genes identified using ABRicate.

L214: "... genes part of pathogenicity islands and..."

Edit accepted and changed.

L251: A more thorough screening of AMR and virulence factor should be performed using curated databases.

We agree with the above statement, and the genes are now confirmed using ABRicate and the manuscript updated accordingly.

L252: "In comparison, ... terrestrial ..." something is missing in the sentence.

The word "terrestrial" was a remnant of an edit. It has been removed.

L257: There is no experimental evidence that the identified genes are indeed involved in AMR. Based on the annotation, it is not obvious to which antibiotic these putative AMR genes would provide resistance to. Providing this information would be necessary.

We agree with this critique. We have changed the wording to reflect the uncertain nature of the resistance genes and discuss the high confidence AMR annotations to include the antibiotic resistance associated with those genes.

L270-272: There is no experimental evidence to support this hypothesis. Other methods of phage detection should be used. For example, the prophage sequences are known, so the authors could induce their lysogenic strains and try to detect by PCR the induced phages in the lysate, after adequate DNase/RNase treatment and proper controls to detect bacterial DNA contamination. Whole phage DNA/RNA extraction can even be done on induction lysates and restriction profiles can be done. The authors cannot conclude that the prophages did not induce upon UV treatment without further experimentation and additional methodological details.

We agree this comment was speculative and have stricken it from the manuscript.

L289: by "complete genome" do you mean closed genome without gaps? If so, why some prophages were found to be split on two or more contigs after comparing with MinION sequencing?

We have edited the text for clarity. The ISS flight genomes are all "scaffold" or "contig" so while we were able to use "complete" genomes from the terrestrial dataset, the flight genomes were all incomplete. This motivated the use of long-read assemblies, which is also now in the text.

L294: Were the results from PhiSpy equivalent to those from PHASTER? Were any discrepancies detected? This should be mentioned in the text. Also: "...confer fitness to various antibiotics". Verify sentence: ... confer resistance or tolerance to various antibiotics maybe?

We have updated the text to indicate where PhiSpy and PHASTER differed and the rational for using

each. In particular, we were encouraged that all the PHASTER “intact” prophages were consistent with PhiSpy. We also edited the sentence to read “associated with putative virulence or antibiotic resistance functions”

L300: 90% nucleotide identity
Edit accepted and changed.

L326-327: Additional experimental details are needed for the phage detection section.
This section has been updated with the details on the mitomycin C and UV experiments.

L329: “... by phenol-chloroform...” ?
Edit accepted and changed.

L332: All gDNA manipulations were performed.
Edit accepted and changed.

All bacterial names should be in italic in the reference section.
Edit accepted and changed.

Reviewer #2 (Remarks to the Author):

The manuscript by Irby and Broddrick sequenced 245 genomes and obtained the metagenome-assembled genomes from bacteria collected in the International Space Station. The authors found the lysogeny prophages are different in spaceflight strains and their terrestrial counterparts. Overall, the study is interesting and the writing is very clear. However, the manuscript has two major concerns: one that the major conclusion is based on the comparison of the sequenced spaceflight strains and the sequenced terrestrial strains. Differences of prophages are common in microbiomes isolated from two different environmental niches on earth, thus this conclusion seems premature without proper control strains. The second is the Program PHASTER and RefSeq Genome Database used to predict prophages have limitations. It is important have to a comprehensive or a combination of different programs to get a more accurate analysis of complete and incomplete prophage in these strains and for the prediction of prophage encoded functional genes.

Regarding the computational pipeline, we have expanded our workflow to include PhiSpy in addition to PHASTER and limiting our intact/complete prophage analysis to those that were consistent with both tools. Reassuringly, all our previously identified intact prophage regions were consistent across both algorithms. We also added AMRFinder to our pipeline, which includes genes from the Comprehensive Antibiotic Resistance Database (CARD), confirmed literature review by database curators, and collaboration with Gen-FS and subject matter experts.

Additionally, we agree that environmental differences in phage is common; however, our primary focus was on the novelty of the prophages in ISS isolates. Our conclusion that the functional content of those prophage regions is correlated to fitness in the built spacecraft environment reflects our agreement that additional work is needed to make more mature claims.

REVIEWERS' COMMENTS

Reviewer #3 (Remarks to the Author):

I commend the authors for making the changes in this revised version. All my concerns have been addressed adequately. I have no additional concerns.